# Mediators of Regional Kidney Perfusion during Surgical Pneumo-Peritoneum Creation and the Risk of Acute Kidney Injury—A Review of Basic Physiology

**DOI:** 10.3390/jcm11102728

**Published:** 2022-05-12

**Authors:** Csaba Kopitkó, László Medve, Tibor Gondos, Karim Magdy Mohamed Soliman, Tibor Fülöp

**Affiliations:** 1Department of Anesthesiology and Intensive Therapy, Uzsoki Teaching Hospital, Semmelweis University, H-1145 Budapest, Hungary; 2Department of Anesthesiology and Intensive Therapy, Markhot Ferenc Teaching Hospital, H-3300 Eger, Hungary; dr.medve.laszlo@weconnect.hu; 3Doctoral School of Pathological Sciences, Semmelweis University, H-1088 Budapest, Hungary; gondos.tibor54@gmail.com; 4Department of Medicine, Division of Nephrology, Medical University of South Carolina, Charleston, SC 29425, USA; soliman@musc.edu (K.M.M.S.); tiborfulop.nephro@gmail.com (T.F.); 5Medicine Service, Ralph H. Johnson VA Medical Center, Charleston, SC 29401, USA

**Keywords:** acute kidney injury, intra-abdominal pressure, oxidative stress, renal cortical blood flow, renal medullar blood flow, renal lymphatic drainage, venous congestion

## Abstract

Acute kidney injury (AKI), especially if recurring, represents a risk factor for future chronic kidney disease. In intensive care units, increased intra-abdominal pressure is well-recognized as a significant contributor to AKI. However, the importance of transiently increased intra-abdominal pressures procedures is less commonly appreciated during laparoscopic surgery, the use of which has rapidly increased over the last few decades. Unlike the well-known autoregulation of the renal cortical circulation, medulla perfusion is modulated via partially independent regulatory mechanisms and strongly impacted by changes in venous and lymphatic pressures. In our review paper, we will provide a comprehensive overview of this evolving topic, covering a broad range from basic pathophysiology up to and including current clinical relevance and examples. Key regulators of oxidative stress such as ischemia-reperfusion injury, the activation of inflammatory response and humoral changes interacting with procedural pneumo-peritoneum formation and AKI risk will be recounted. Moreover, we present an in-depth review of the interaction of pneumo-peritoneum formation with general anesthetic agents and animal models of congestive heart failure. A better understanding of the relationship between pneumo-peritoneum formation and renal perfusion will support basic and clinical research, leading to improved clinical care and collaboration among specialists.

## 1. Introduction

Patients with postoperative acute kidney injury (AKI) show significantly inferior survival rates compared to patients with normal kidney function [1]. Moreover, recurrent AKI represents a risk factor for future decline of kidney function [2,3,4] The incidence of AKI falls between 2 and 41% after major intra-abdominal surgery, depending on the type of surgery and the criteria employed for the diagnosis of AKI [1,5,6,7,8,9,10,11].

While practicing nephrologists pay close attention to intake and output balances as well as intra-operative blood pressures, the role of surgical methods in the development of postoperative AKI is underappreciated. Non-surgeons may find operating room records difficult to read, and in the time-pressed environment of clinical practice it is critical for nephrologists to focus on factors influencing intraoperative renal perfusion. Laparoscopic surgery is gaining ground due to decreased postoperative recovery times and perceived lower surgical burden but it operates at inflated gas pressures nominally classified as intra-abdominal hypertension stage I or II [12,13].

The trans-endothelial fluid shift is governed by the hydrostatic and colloid osmotic forces of both side of the glyco-calyx according to Starling’s principle [14]. The net ultrafiltration pressure is derived from the opposing forces of the glomerular blood hydrostatic pressure (approximately 55 mmHg under physiological condition in humans) vs. the combinations of capsular hydrostatic pressure (about 15 mmHg) and blood colloid osmotic pressure (about 30 mmHg) [15]. With an assumption of unchanged capillary permeability and surface area and an ultra-filtrate that is practically protein-free without colloid osmotic pressure, the net ultrafiltration pressure calculates to around 10 mmHg only, a surprisingly low number.

Increased intra-abdominal pressure (IAP) can result in the elevation of hydrostatic pressures within both the Bowman’s capsule and in the renal interstitium caused by the impairment of renal venous drainage, lymphatic drainage or both, but its effect can vary on a case-by-case basis [16,17]. With the establishment of laparoscopic pneumo-peritoneum, the intra-abdominal pressures reach 15 mmHg and laparoscopic surgeries may be more prolonged compared with conventional open approaches [12,13]. Alternative techniques to visualizing the operative field with retroperitoneal insufflation or gasless laparoscopy by abdominal lifting are published, but not extensively utilized.

The aim of this review article is to discuss the new and emerging basic scientific data and place this knowledge in the proper historical context of older and partly forgotten information, in order to refocus attention on the importance of effective perioperative renal perfusion to interpret AKI risk.

## 2. Arterial Blood Supply and Renal Blood Flow

The kidneys alone account for approximately 20% of cardiac output uptake at rest, a figure that has been remarkably preserved across mammalian species. As is widely known, renal autoregulation keeps the renal plasma flow (RPF) stable under physiological conditions, when the mean arterial pressure (MAP) is maintained between 80 and 160 mmHg, mirroring the balance between the afferent and efferent arteriolar tones in the cortex [18,19]. The renal blood flow (RBF) has a significant circadian oscillation: the peak value can reach 150% of the nadir RBF [16,17]. The intra-renal distribution of blood flow is extremely disproportionate: approximately 90% perfuses the cortex only, and only the remaining 10% provides for the metabolic needs of the medulla [18,20]. An elevated IAP caused by pneumo-peritoneum is expected to result in a diminished renal perfusion to 40–50% of its baseline level [21]. On the other hand, any alteration of the medullary flow is secondary, as it depends mainly on the outflow emerging from the cortical vascular bed.

The main contributors of filtration gradient are the hydrostatic and osmotic forces, but fine tuning is exerted by some other mechanisms, such as the myogenic response (renal arterial vasodilatation in an acute increase of IAP), tubulo-glomerular feedback, inflammatory and other humoral factors, which will be detailed later. The myogenic response serves as the most immediate regulatory process, which can show remarkable inter-individual variability based on animal studies [22]. 

Most of our knowledge regarding the regulation of intra-renal regional flow is derived from mammalian animal models. Most animal experiments were conducted in pigs and rats, and occasionally dogs or rabbits [23,24]. In rats, and even in humans, approximately 10% of the juxtamedullary nephrons encompass pre-existing shunts between afferent and efferent arterioles, modulating the number of functioning nephrons (Figure 1) [18,25]. Furthermore, intraluminal valves and so-called endothelial cushions have been described at the connection points of interlobular and afferent arterioles [18,26,27,28]. 

These anatomical structures with the capacity to alter the vascular tone can contribute to the global and regional inflow to the medullary vascular bed. The vasa recta deriving from juxtaglomerular nephrons surrounds the proximal convoluted tubules when tracking the descending part of Henle’s loop, then distributes into capillaries, which are collected in venules running around the distal convoluted tubules [29,30]. According to the anatomical scenario described above, the tip of Henle’s loop is the location most sensitive to hypoxia. Aglomerular arteries were described in a number of cases in kidney corrosion cast preparations, whereas these originate from afferent arterioles and end in vasa recta [18]. The aglomerular vessels can play a role in adjusting the medullar interstitial milieu. In addition, definite shunts are detected in the vasa recta itself, which in an open state can profoundly impair medullary blood flow, especially at the tip of Henle’s loop. Descending vasa recta form bundles and the arterioles placed in the central part of the bundle reach the tip of the papilla [20]. Only 20% of the blood flow passes into the inner stripe of the medulla [31]. A meticulous regulation of medullary blood flow is essential to maintain adequate kidney function, as an unregulated increase of blood flow would wash out the tubulo-interstitial osmotic gradient and eliminate the urinary concentrating capacity. An excessive decline of blood flow would result in papillary necrosis, since the partial pressure of oxygen in the medulla is only 20–40% of that in the cortical tissue [20]. 

### 2.1. Renal Medullary Circulation

The medullary vascular bed is sequentially connected to the cortical outflow. The relationship between renal and medullary blood flow seems to be curvilinear, while the cortical blood supply remains stable [18,32]. The muscular layer of efferent arterioles is gradually replaced with pericytes as the vessels branch into the vasa recta (Figure 2) [18]. These cells are found a certain distance apart from each other, with their anatomical distances gradually increasing along the descending vasa recta in the inner layer of the medulla [20]. Near the tip of Henle’s loop, where the continuity of endothelial cells (seen in the descending vasa recta) transmits into the ascending vasa recta’s fenestrated endothelium, pericytes are undetectable. Pericytes are cells with phenotypes remarkably similar to vascular smooth muscles, having claw-like processes surrounding the external surface of endothelial cells and being able to contract both tangentially and circumferentially. The pivotal role of renal pericytes has been rediscovered recently: these cells are able to modulate by 10–30% the diameters of the vessels they wrapped around [33]. The medullary thick ascending limb, the collecting duct and the vasa recta are in close proximity to each other, while the adjacent pericytes provide the vascular tone mediation of the direct tubulo-vascular communication [20].

Pericytes are under the control of multiple agents derived from (1) the nervous system (acetylcholine via muscarinic receptors, noradrenalin via α_1_-receptors), (2) circulating vasoactive agents (vasopressin via V_1 a_-receptors), and (3) locally released agents (angiotensin II via AT_1_-receptors, endothelin-1 via ET_A_-receptors, adenosine triphosphate and uridine triphosphate via P_2_-receptors) as vasoconstrictors [20,28,33,34,35,36,37]. Three of these factors (acetylcholine, angiotensin and adenosine triphosphate) can also lead to vasodilatation in an NO-mediated manner [20,28,33,34,35,36,37]. Some agents, such as PGE_2_, PGI_2_, vasopressin via V_2_-receptors and medullipin II also result in vasodilatation [20,33,34,35,38,39]. The nervous system and the systemic circulating agents act mostly in the outer medulla, whereas control local factors predominate in blood flow in the inner medulla [20,34]. Besides the pericytes, the total RBF is also under predominant neural control: an increase in the renal sympathetic nerve activity is associated with a reduced glomerular filtration rate (GFR) and a narrowed auto-regulatory range [40,41]. Renal sympathetic activity has an impact on renin release (via activating β_1_-adrenoreceptors in the juxtaglomerular granular cells), sodium and water reabsorption (by upregulating Na^+^-H^+^ exchanger isoform 3 via α_1_-adrenoreceptor mediated mechanism), and RBF itself [42,43]. 

Fluid administration is usually the first attempt to restore the systemic hemodynamics in clinical practice. It is an effective tool for providing adequate organ perfusion in many cases, but it can also lead to fluid overload with commensurate organ dysfunction in multiple locations. In the kidneys, acute volume expansion suspends the auto-regulatory capacity of medullary vessels in rat models, while cortical circulation remains unaffected [32,44]. Cardiac output monitoring and goal-directed fluid therapy are essential as supported by experiments conducted in porcine models, but over-hydration can cause a decline in urinary output [34,45]. 

Additionally, intravenous fluid administration may result in excess hemo-dilution in the renal vascular bed. The hematocrit in vasa recta vessels is only one-third of that measured in the systemic circulation in volume-expanded rats [32,44,46]. This hematocrit remains stable regardless of the alterations of perfusion pressure. The velocity of red blood cells is increased, therefore their transit time is shortened to at least 50% according to studies conducted with radioactively labelled albumin and red blood cells [32,44,46]. As a consequence, the oxygenation of the medulla can worsen, since lower contents of red blood cells spend less time in the capillaries, allowing less time for gas exchange between red blood cells and the renal tissue.

### 2.2. The Effects of Pneumoperitoneum on Renal Blood Flow

In the past few decades, laparoscopy has brought a revolutionary improvement to the field of surgical techniques. After the induction of anesthesia, the abdominal wall is elevated manually, and the abdominal cavity is insufflated with carbon dioxide (CO_2_) via a Veress needle [47]. This needle was introduced into clinical practice in 1938 by Dr. János Veress, a Hungarian internist. The Veress needle was initially developed to ensure a safer pneumothorax creation in order to collapse lung infected by *Mycobacterium tuberculosis* for healing. It is to be noted that the possibility of abdominal puncture is mentioned in the original publication. The device is made of an outer needle having sharp bevel and a dull-tipped, spring-loaded inner stylet is positioned in it. The inner stylet is kept in place and pushed back when resistance is felt, but after passing the parietal layer of peritoneum into the abdominal cavity, the resistance disappears, and the protruding blunt tip protects the viscera from accidental perforation. Contrary to this closed approach, an open technique has also been described [48]. 

Surgical pneumo-peritoneum formation lowers RBF by about 40% and results in a decrease in urine output and creatinine clearance [21,49]. Both the level of insufflating pressure and the duration of insufflation can impact the GFR and urinary sodium excretion even in cases where MAP remains unchanged [21]. Applying 7 mmHg of pneumo-peritoneum decreased the urine flow and fractional urinary sodium excretion in rats, which was more pronounced when the duration of insufflation time was raised from 30 to 60 min. The net ultrafiltration pressure was reported as 14 mmHg, which is very similar to the one calculated in humans (10 mmHg) [50]. The reduction of RPF was much more noticeable when IAP raised to 14 mmHg, but it became independent from the length of the insufflation interval [21]. These effects were less marked when, instead of CO_2_, helium or argon was insufflated intra-peritoneally [51,52,53,54,55].

Further insights can be obtained from porcine models by examining the differences in flow dynamics between the medulla and the cortex upon increased intra-abdominal pressure. In a porcine model, the initial renal cortical blood flow was 5.5 times greater (50 ± 18 vs. 9 ± 3 mL × min^−1^ × 100 g^−1^ (tissue)) than the medullary blood flow once the abdomen was insufflated and a laser-Doppler flowmetry probe was introduced into the renal parenchyma [56]. With a progressive increase of IAP from 0 to 40 mmHg, the cortical flow decreased exponentially, but the medullary flow increased until an IAP of 20 mmHg was reached and declined thereafter once insufflation pressure escalated further (Figure 3). At 15 mmHg of IAP, the cortical and medullary blood flows were effectively equalized. A relatively low rate of medullary blood flow helps to maintain a high osmotic gradient and low interstitial hydrostatic pressure; therefore, these changes seem to be deleterious for kidney function [29,30]. It is conflicting that the RBF data in this study are far lower than in previously published studies on the hemodynamic distribution of intra-renal circulation (700 mL × min^−1^ × 100 g^−1^ (tissue) for renal cortex, 300 mL × min^−1^ × 100 g^−1^ (tissue) for the tissue near the cortico-medullary junction, 200 mL × min^−1^ × 100 g^−1^ (tissue) for the inner stripe of the outer medulla and 50–100 mL × min^−1^ × 100 g^−1^ (tissue) for the inner medulla) without any reference to the types of animal models [16,18,29,32]. Looking at these data, we can conclude that the pneumo-peritoneum depending on its duration markedly abolishes renal cortical perfusion and increases the medullary blood flow at the level of pressure used in everyday clinical practice. Increased medullary perfusion can lead to a decreased concentrating ability in the kidney.

### 2.3. The Direct Effect of CO_2_ on Renal Vasculature

The direct effect of CO_2_ on renal circulation has been investigated extensively [57,58,59]. In ten mildly dehydrated dogs, RBF was directly measured in multiple ranges (<30, 30–50, 50–70, 70–100 and >100 mmHg) of arterial partial CO_2_ with a gradual increase of arterial CO_2_ [57]. The authors reported an 11% decrease in RBF over PaCO_2_ of 70 mmHg, and a further 7% decrease over PaCO_2_ of 100 mmHg compared with the RBF between PaCO_2_ of 30–35 mmHg. This effect was abolished by pharmaceutical renal denervation or after the administration of mannitol. However, if we recalculate the results taking into consideration the individual changes from the data of the original article, we can discover other interesting details. Elevated PaCO_2_ has a heterogeneous effect on renal circulation. While the mean value of RBF moved downward, its individual value was increased by 6% in three dogs over PaCO_2_ of 70 mmHg, and by 15% in one dog over PaCO_2_ of 100 mmHg. The decrease was consequently more serious in the remaining animals: an 18% drop over PaCO_2_ of 70 mmHg and a 24% fall over PaCO_2_ of 100 mmHg. Renal denervation was performed in five dogs: two of them showed increased RBF over PaCO_2_ of 70 mmHg compared with baseline, but each of the five showed a 4% improvement in comparison with the data produced by the first group. CO_2_ restricts RBF unpredictably with results differing from animal to animal, but it is only at the supra-normal arterial CO_2_ level that this has any clinical significance.

Pneumo-peritoneum-associated metabolic acidosis was attenuated in a swine model, when helium was applied instead of CO_2_ [60]. This phenomenon draws our attention to the direct vaso-dilatory and indirect (through respiratory acidosis) renal effects of CO_2_, although no differences were reported in urine output between pneumo-peritoneum created by CO_2_ and argon in swine models [61]. 

## 3. Venous Drainage and Congestion

A separate, but equally important issue is the consideration of venous pressures impacting renal perfusion. The fact that increased intraperitoneal pressure is strongly associated with the decrease of urine output and the deterioration of the excretory function while MAP remains unchanged implies the possible role of venous congestion in the development of perioperative AKI [21]. The splanchnic organs serve as a reservoir of 25% of the total blood volume under normal physiological conditions [62,63,64]. The contained blood volume has a hematocrit level over 70%. It can be pushed into the systemic circulation when the sympathetic nervous system activated resulting in a significant intravascular volume expanding effect when the red blood cells are diluted up to physiologic level [62]. Augmented venous return to the right atrium may lead to increased cardiac output, but on the other hand increased IAP can cause renal venous congestion. Further modulation can be provided by the activity of the renin-angiotensin-aldosterone system (RAAS) [18,41]. Several other humoral and paracrine factors, including the participation of atrial natriuretic peptide, endothelins, nitric oxide and prostaglandins, can also influence shifting blood [18,41,62].

The first findings were taken from a dog heart-lung-kidney model studied by Winton and his coworkers [65]. The applied venous pressure was 24 mmHg, which had deleterious consequences identical to a 15 mmHg drop of arterial pressure. The deterioration of renal function to a certain amount could be reverted by establishing lower-than-normal air pressure conditions in a chamber surrounding the kidney. The relationship between renal venous and interstitial pressures and its importance in urine production have been studied in detail in the next two studies, subsequent to Winton’s original publication [66,67,68,69,70]. These investigations revealed that urine output starts to decline once renal venous pressure reaches 15–20 mmHg [69]. On the other hand, the influence of elevated venous pressure could be counterbalanced by increasing MAP [68]. Renal interstitial pressure rose during this procedure and RBF decreased just after a 40 mmHg of renal venous pressure was reached. The pattern of decreased urine output showed significant heterogeneity: it can be delayed by 15–20 min or even omitted, despite the three- to-four-fold increase in venous pressure [66]. The authors concluded that both insufficient MAP and elevated venous pressure can worsen kidney function, but kidney function can be maintained as long renal perfusion pressure is maintained.

Since the kidneys are encapsulated organs, gaining even a small interstitial volume can result in a disproportionately high rise of intra-parenchymal pressure leading to what could be termed a “renal compartment syndrome” [71]. This underlines the importance of venous congestion, which has remained underemphasized until recently [70,72,73,74,75]. Once congestion occurs, an interstitial edema is formed.

### 3.1. The Effect of Pneumoperitoneum on the Blood Flow in the Inferior Vena Cava

Placing ultrasound probes around the retroperitoneal vessels afforded further exploration of the quantitative relationship between insufflation pressures and visceral blood flow rates. In a rat model of experimental laparoscopy, this approach demonstrated a gradual decrease of average blood flow [76]. The average blood flow in the inferior vena cava dropped to 7% of the baseline caval vein flow when the abdomen insufflated to 10 mmHg and a further decrease (to 3% of baseline) was observed at 15 mmHg, while aortic flow was relatively maintained (54% and 40%, respectively). As a conflicting result, the renal venous blood flow was reported to slightly increase below IAP of 15 mmHg, but decreased to 50–75% of baseline above IAP of 15 mmHg in a porcine model [61]. 

### 3.2. The Backward Effect of Increased Renal Venous Pressure

Evidently, any obstruction in renal outflow can limit the arterial inflow and consequently the renal perfusion and kidney function. Unexpectedly, hampered venous outflow leads to a greater damage in RBF than the cessation of arterial inflow. To investigate this effect, renal venous pressure was raised experimentally in a porcine model [77]. The vessel loop around the renal veins was maintained for two hours. By the end of this two-hour period, both the renal artery blood flow index and GFR were reduced, and a modest proteinuria developed.

Similar results were reported in a two-hour retroperitoneal CO_2_ insufflation of 10 mmHg in rabbits [78]. The renal artery flow rate was slightly decreased, reaching about 75% of baseline after 4 h. The venous flow rate reached 75% at 2 h (two hours earlier than the arterial flow rate) and declined further to 50% of baseline at the 4 h mark. The change of blood flow evoked by laparoscopy in the renal arterial and venous systems are similar in tendency, but greatly different in their capacity. The effects of renal venous congestion were discussed in detail in our previous publications [70,79,80]. 

### 3.3. The Role of the Renal Capsule

Theoretically, when the drop in renal venous blood flow is disproportionately greater than the reduction of RBF, a significant amount of fluid must be retained in the kidney tissue. To our knowledge, it has not been investigated where the excess retained fluids were to be drained. It is unlikely to be the urinary direction, since its output kinetics are similar to that of the venous flow rate [72,81]. The additional interaction with the lymphatic flow is discussed below. Actually, the severely diminished venous outflow should provide drainage for the slightly increased arterial input, whereas the decreased urinary output does not help to alleviate the exponentially increased pressure inside the tight renal capsule. The importance of the kidneys being encapsulated organs was further highlighted by past observation that removing the layer of renal capsule in seriously injured patients prevented the development of acute kidney injury in human reports [82]. 

The interstitium is a biologically active space, with albumin accounting for most of the interstitial oncotic pressure. Extravascular albumin accumulates gradually, increasing from the cortex to the tip of papilla; therefore, the presence of fluid overload exacerbates regional interstitial edema formation and can culminate in severe increment in the pressure of the inner stripe of the medulla [46].

### 3.4. The Determinants of Renal Venous Pressure

Moving on to the next interaction, the interference of IAP and central venous pressure (CVP) was revealed in a study conducted in rats [83]. The establishment of pneumo-peritoneum by nitrogen (insufflation of 20 mmHg for 4 h) resulted in an elevated CVP and a greatly decreased (<50%) RBF. Primarily metabolic acidosis was developed, which was soon complicated by respiratory acidosis. Although hemodynamics returned to its basic level after the release of abdominal pressure, AKI occurred and lung tissue damage became evident in the histology. Similar results were found in a study conducted in pigs [60]. 

In addition, elevated central venous pressures can impede the emptying of lymph from the thoracic duct leading to significant backward effects and a potential escalation to intra-renal edema [84,85,86,87,88]. This theory is seemingly in contradiction with the experiment where venous pressure was registered both above (internal jugular vein) and below (iliac vein) the diaphragm of swine [89]. RBF decreased gradually when the pressure of nitrogen pneumo-peritoneum was raised from 5 to 25 mmHg and the iliac vein pressure moved in parallel with these changes, but the CVP did not.

## 4. Lymphatic Drainage of Renal Tissues

Renal interstitial pressure changes synchronously to renal perfusion pressure [32]. The net pressure difference guiding the filtration in the glomerulus is rigorously regulated, and moves around 10 mmHg under physiologic conditions [15,90]. The mean capillary pressure in the medulla oscillates in a slightly lower range, at about 7 mmHg [15,91]. The renal parenchyma is primarily drained by the urine conducting and renal venous system. The renal lymphatic system can also contribute to renal compartment syndrome, but to our knowledge the functional changes in the lymphatic flow was not investigated in connection with pneumo-peritoneum. Unfortunately, the experimental data for renal lymphatics divert markedly between animal species: the lymphatic outflow is published as 1.5–3.0 mL/h in female sheep, but can be as high as 150 mL/h in dogs [92,93]. Such divergent observations make it impossible to synchronize these findings with human conditions, but the amount of lymphatics generated is estimated to be similar to the amount of urine output [94]. The anatomical and physiological background of this wide range will be highlighted below.

### 4.1. The Hilar and the Cortical Route

The cortical lymphatic capillaries begin near the Bowman’s capsule and the blind ends of the medullary capillaries are observed in the sub-mucosal layer of the papilla in humans (Figure 4) [84,95]. By contrast, several publications argue against the existence of medullary lymphatics. Most authors who are in favor of their existence restrict their origin to the outer, fluid-rich medulla and reject the theory of a deeper-layer origin [84,96,97,98]. One possible explanation is that significant interspecies and inter-individual differences, e.g., lympho-venous communication, were demonstrated in many animal autopsies, but not in humans [84,99]. The main route for draining renal lymph is the hilar path under physiologic conditions (the flow is 4–8 times greater through the hilar than the renal direction). The channels perforating the capsule of the kidney bear a far lower importance.

The two systems are connected through communicating tubules (Figure 4) [84,100]. The hilar lymph can be diverged to the capsular system, e.g., after ureteric obstruction [101]. The electrolyte concentration of the hilar lymph is almost identical to the electrolyte concentration of the plasma, implying that the cortical lymph is also drained in the hilar direction [84,102]. The source of lymph may be tracked by a mixture of labeled glucose and mannitol. Glucose is reabsorbed in proximal tubules, while mannitol is filtrated only. Accordingly, a higher glucose/mannitol ratio in the renal lymph would suggest the medullary origin of the lymphatic fluid under physiologic circumstances. During donor nephrectomy, lymphatic vessels are transected, but start to regenerate within 7 days, leading to intactly functioning lymphatics 2–3 weeks later [103]. The activity and completeness of lymphangio-genesis can be associated with a lower rejection rate [103,104]. 

### 4.2. The Microanatomy of Lymphatic Capillaries

During lymph generation, the fluid enters the lymphatic capillaries either passively through the gaps between the “button-like” (i.e., tight connecting points certain distances apart) intercellular junctions in contrast to the “zipper-like” junctions between the endothelial cells of the blood vessels, or via the active trans-cellular uptake across the endothelial cells [84,103]. The blunt openings of the vessels are tethered to the surrounding matrix of the tissue, with this anchoring preventing the collapse of the ducts [84,105]. 

### 4.3. Lymph Moving Forces

Unlike amphibians, which possess so-called lymphatic hearts (up to 15 pairs), in mammals lymph is mostly propelled by the movement of surrounding tissues such as muscle contractions and bowel peristalsis or by passive forces, with pressure differences, e.g., derived from the respiratory cycle [98]. The one-way valves found in lymphatic vessels ensure a unidirectional flow, which is generated by the active contractions of smooth muscles in their wall [101]. These valves are formed from the overlapping junctions of endothelial cells at the beginning of the ducts, but in the larger lymphatic canals true traditional valves can also be observed [105].

Renal vein compression increases lymphatic pressure within five minutes [19,84,93]. In contrast, the occlusion of ureters exerts a more gradual effect, whereas the lymphatic drainage system can cope with the inflow to the renal compartment only within a certain range (Figure 5). As Rohn et al. nicely demonstrated in dogs, when the venous pressure rose to 20–25 mmHg, lymphatic resistance decreased, lymphatic driving pressure increased and lymphatic flow was augmented [93]. The new steady state was reached in about 30–45 min. The lymphatic pressure-flow curve was shifted to the right when renal venous pressure was elevated, a phenomenon which could be special for the kidneys. This mechanism can serve as a safety release valve to escape from parenchymal pressure elevations.

There were individual differences detected in lymphatic flow rates and the renal pressures at which the lymphatic flow started to decline. Inter-individual variations in the development of the perforating system or medullary drainage can explain the different consequences of the elevated renal venous pressure in dogs mentioned above [66]. Further possible compensatory processes are the diversion of lymph from hilar to capsular vessels, which was observed in dogs after three days of ureteric compression, and lymphangio-genesis taking place in a larger time-frame [84,101,105,106,107,108,109]. 

### 4.4. The Turnover of Albumin, the Key Element of the Interstitium

5–25% of the total peri-tubular vascular endothelial surface is occupied by large (0.04–0.05 µm) pores [46]. Albumin is a flexible molecule, 3.8 nm in diameter and 15 nm in length, but the split-size of a rectangular pore has been reported as small as 35 Å (3.5 nm) in cross-section [110]. It is this mobility that makes albumin capable of transmission easily through capillary pores, while the larger size globulins are retained intra-luminarly. The protein concentration of renal lymphatic fluid is 43% of plasma in sheep, but the proportion of albumin is higher than in serum (albumin/globulin ratio is 1.3 vs. 0.69) [46,84,92]. The extravascular albumin pool is important for the maintenance of oncotic pressure in the interstitium: as mentioned earlier, its concentration is at least twofold in plasma at the tip of the papilla as compared to peripheral blood [46].

After intravascular injection, fluorescein-labelled albumin can be detected as early as 40 s in the extravascular space of rat kidneys [46]. The transmission of radioactively labelled albumin into the renal lymph takes about 2 h, but 85% of the equilibrium is realized within 3 min in the papilla and 1 min in the cortex [46]. This fact can be explained by a significant reuptake of albumin on the venous side. Albumin exchanges with a short turnover time in kidneys, but about 30% resides in the slower exchange compartment, supposedly in the extravascular space of the medulla [46,84].

## 5. Humoral Factors

### 5.1. The Renin-Angiotensin-Aldosterone System

The renin-angiotensin-aldosterone system (RAAS) is one of the main contributors of renal macro- and micro-hemodynamics. Its widely known parts are renin, angiotensinogen, the angiotensin converting enzyme, angiotensin II, and aldosterone. Some of the other fragments have no biological effects (angiotensin I), or have no effects in humans, but have in rodents, dogs or both (angiotensin IV, angiotensin 1–7n’ hepta-peptide cleaved from angiotensin I, angiotensin A octa-peptide generated from angiotensin II by the decarboxylation of asparagine) [111]. The main result of RAAS activation is the elevation of blood pressure via either direct vasoconstriction or salt and water retention. The renal effects of angiotensin II and angiotensin III are independent from their serum levels because these agents are formed locally at about a thousand times greater amount [111]. Moreover, angiotensin II is a powerful trigger of aldosterone secretion.

Angiotensin receptors are classified as angiotensin receptor type 1 (AT_1_, which has two subtypes in rodents: AT_1a_ and AT_1b_) and angiotensin receptor type 2 (AT_2_) [111]. AT_1_ receptors are expressed in multiple sites in the kidney, while AT_2_ receptors are found mainly in fetal and newborn mammalian kidneys. In adult mammals, AT_2_ receptors are limited to glomerular mesangial cells, to the pre-glomerular arcuate and interlobular arteries, but can be upregulated in sodium-depleted states.

The decrease in RBF is AT_1_-receptor-mediated, but efferent arterioles are more susceptible and thus the effective glomerular filtration pressure can be maintained. Contrary to cortical vasoconstriction, angiotensin II causes arterial vasodilatation in the renal medulla. AT_2_ receptors trigger nitric oxide, bradykinin and cyclic guanosine monophosphate production, exerting the opposite effects when compared to AT_1_ receptors. The result is decreased sodium excretion at low doses of angiotensin II, which reverts to blood-pressure-related natriuresis and diuresis, also known as ‘pressure-diuresis’.

The two main causes of renal venous congestion in human pathophysiology are heart failure and fluid overload. These can lead to completely different renal effects depending on the activity of RAAS [70]. An increased sensitivity to angiotensin II was reported after the surgical denervation of the kidneys in sheep [40]. 

### 5.2. The Tubuloglomerular Feedback

The myogenic mechanism of afferent arterioles and the tubulo-glomerular feedback are the main contributors of RBF besides sympathetic innervation [19]. A so-called third mechanism (3M) also exists, representing vaso-constrictive forces acting on afferent glomerular arteries with slower reaction time. The physiological background of 3M is uncertain, but some argument supports the possible role of RAAS [112]. The concept of tubulo-glomerular feedback describes the connection between the distal convoluted tubule and the macula densa. This anatomical juxtaposition provides information about the individual intra-tubular fluid’s sodium concentration and osmolality and influences the glomerular inflow by regulating renin secretion. This mechanism is less effective when angiotensin II levels are suppressed, but it is exceedingly responsive if the RAAS is triggered [19]. Serum aldosterone levels were shown to be raised by 40% during the application of intraperitoneal and up to 90% during retroperitoneal insufflation in swine [113]. 

### 5.3. Nitric Oxide

Various stimuli, such as shear stress, thrombin or bradykinin can elicit the release of vasoactive agents [19]. One of the most potent products of this sort is nitric oxide, which is formed by neural, constitutive and inducible endothelial enzymes. The blockage of inducible or endothelial nitric oxide synthases causes a 25–40% elevation in the vascular resistance of the kidney [19,20,114]. By contrast, the adverse effects of pneumo-peritoneum on the RPF and GFR (measured by the clearance of inulin and para-amino-hippuric acid) can be ameliorated by pretreatment with a nitric oxide donor. The effect of pneumo-peritoneum on RPF and GFR is aggravated with endothelin-B antagonists or by blocking nitric oxide production in rats [21,115]. The activation of endothelin-B receptors elicits nitric oxide and the release of prostaglandin from the endothelium [115]. The establishment of pneumo-peritoneum itself reduces the vasodilator S-nitroso-hemoglobin concentration, which can be reversed by inhaling S-nitrosylating agent ethyl nitrite [116]. 

In rat studies, the administration of nitric oxide alleviates the renal effect of pneumo-peritoneum at a higher (14 mmHg compared to 7 and 10 mmHg) pressure by about 40%, as can be gleaned from a bar chart in Bishara et al., while the nitric oxide synthase inhibitor aggravates it by approximately 50% [117,118]. 

Urinary nitric oxide metabolites are increased in compensated, but not in decompensated, chronic heart failure induced by establishing an artificial aorto-caval fistula in rats, implicating that this mechanism becomes exhausted with advancing heart failure [118,119]. Regrettably, the anatomical relationship between the aorto-caval fistula and renal vessels (whether infra- or suprarenal) was not communicated in the latter publications [118,119]. The deterioration of RPF was less pronounced in the compensated heart failure group than in the control group at 10 and 14 mmHg. The RPF and GFR were dissociated from each other: when the peritoneum insufflated to the pressure of 7 mmHg in rats with compensated heart failure, a minor increase of GFR was detected in spite of the reduced RPF.

The administration of a nitric oxide synthase inhibitor eliminated the beneficial effect and RPF became worse than in the control group with attenuated parallel changes in GFR [118,119]. Hyper-perfusion and a proportionally (about 60%) increased GFR were observed in the control group after the termination of insufflation. Hyper-perfusion was mainly abolished in the compensated heart failure group, but only a slightly lower GFR was observed compared to the values produced by the control group. Blocking the nitric oxide synthase resulted in the opposite effect: higher RPF and lower GFR compared to the compensated heart failure group, but each parameter was lower than in the control group. To summarize these findings, both RPF and GFR showed less marked changes during pneumo-peritoneum in rats with compensated heart failure when results were compared with those of the control animals. According to these results, in compensated heart failure, a pneumo-peritoneum of 7 mmHg is the most favorable scenario. Minimizing the concentration of nitric oxide led to the loss of kidney function during insufflation and similarly in the recovery phase. The dissociation of RPF and GFR can perhaps be explained by the fact that nitric oxide serves as a more potent vasodilator in efferent than in afferent arterioles, which results in a diminished filtration pressure in the glomeruli.

Some experimental results suggest that the increased partial pressure of CO_2_ in the blood can diminish the effects of nitric oxide [116].

### 5.4. The Impact of Obstructive Jaundice

Surprisingly, acute obstructive jaundice was demonstrated to have protective effects against AKI during laparoscopy; however, both bilirubin and biliverdin have antioxidant properties [120,121]. GFR and RPF were found to be lower in rats four days after the ligation of the common bile duct. When pneumo-peritoneum was established, both parameters reached or even exceeded those of the control group at two specific barometric pressures (10 and 14 mmHg for 45 min). No similar results were detected in rats with chronic cirrhosis. Obstructive jaundice induces myocardial dysfunction, which is well-known to be associated with elevated levels of atrial and brain natriuretic peptides and an increased amount of nitric oxide metabolism products in the urine [120]. The urinary concentration of cyclic guanosine monophosphate shows similar changes in GFR and RPF. Cyclic guanosine monophosphate has vasodilatory and natriuretic properties, but the mechanism behind this phenomenon needs further investigation.

## 6. Ischemic-Reperfusion Injury

In certain types of surgery, a temporary cessation of renal perfusion is needed, the consequences of which only add to the detrimental effects of elevated IAP. Perhaps less well-known, the effect of the cessation of venous outflow is similar to that of the arterial occlusion and reaches its peak early. In mice experiments, clamping the renal vein resulted in a greater medullary necrotic area in comparison to the clamping of the renal artery for 30 min (44 vs. 28%) [122]. Extending the clamping period to 45 min, the medullary effects of arterial obliteration grew significantly (77%), while the cessation of venous flow was associated with no further consequences (46%). The compression of the whole pedicle led to a lesion territory of 50% of the whole medulla irrespective of the duration of clamping.

The cortical damage was less serious: about 10% after 30 min for both arterial and venous clampings [122]. In the case of a longer occlusion (45 min), the necrosis increased to fivefold the size of the original lesion zone, but only to 14% when venous clamping was applied. Pedicular bracing resulted in a cortical injury of 8 and 21%, respectively. A 10-min-long cardiac arrest was far less detrimental (4% cortical and 23% medullary necrosis). Summarizing these findings, we can conclude that venous occlusion is actually less tolerable than arterial clamping and that venous occlusion affects the medullary area primarily. In case of the cessation of the arterial flow, significant differences can be detected between the 30 and 45 min period in either cortical or medullary impairment.

During the ischemic period, the enzyme xantin dehydrogenase is irreversibly converted into xantin oxidase in a pressure-related manner during poor tissue oxygenation [123,124,125]. This transformation can be inhibited by the administration of sodium tungstate or tungsto-phosphoric acid, both binding competitively to the active sites of cleaving phosphatases [124]. Ischemia also induces controlled cell-death mechanisms, including necroptosis, mitochondrial permeability transition-mediated regulated necrosis, parthanatos, ferroptosis and pyroptosis [126]. 

### 6.1. The No-Reflow Phenomenon

After a short no-flow or low-flow ischemia, perfusion can be normal or slightly increased, but the very frequent no-reflow phenomenon can be observed during the reperfusion period [127]. This process has been widely investigated in the case of coronary artery interventions. Several mechanisms are hypothesized in its background: (1) microvascular compression due to endothelial necrosis and interstitial swelling, (2) impairment of endothelial-dependent vasodilatation due to deficient NO-production, (3) microvascular plugging with neutrophiles and thrombocytes, (4) large amounts of oxygen leading to the formation of reactive oxygen and reactive nitrogen species.

The ferrous (II)–ferric (III) transition of iron plays a central role in the generation of ROS (Haber-Weiss chain, and the part known as Fenton-reactions) [128]. The product is hydroxyl radical, which is the most toxic agent of ROS [129]. It has a short half-life time (nanoseconds), acts locally in its place of generation, and does not diffuse further, but there is no enzymatic protection against it in humans. Anorganic ROS react with the lipid, protein and nucleic acid structures of the cells producing organic free radicals (half-life time is minutes) or by destroying these molecules [128]. Further reactions with nitric oxide create peroxy-nitrite ions and peroxy-nitrous acid (half-life times are milliseconds), which are cardinal in the development of an ischemic-reperfusion injury [130]. 

### 6.2. Protective Molecular Mechanisms

The activities of superoxide-dismutase, catalase and glutathione peroxidase enzymes with or without scavenger molecules provide the main protective mechanism against reactive oxygen species and lipid-peroxidation products [131]. Superoxide-dismutase activity increases proportionately to the insufflation pressure in rat erythrocytes: it was significantly lower at 5 and at 10 mmHg than in both the control and the sham-operated animals [132]. Superoxide-dismutase activity was higher in sham-operated rats than in rats with pneumo-peritoneum of 15 mmHg. The activity of protective enzymes can be stimulated by ischemic pre- or post-conditioning [133,134]. Both two cycles of a 2.5-min insufflation and a single cycle of 5 min increased the activities of superoxide-dismutase, myeloperoxidase, and attenuated the rise of malondialdehyde (marker of lipid peroxidation) after a 60-min pneumo-peritoneum of 10 mmHg in rats [133,135]. Parallel to these changes, no effect was detected on the inflammatory-response-associated tumor necrosis factor α (TNF-α) levels. It was previously demonstrated that nitric oxide can exert a protective effect on renal circulation, which shows significant heterogeneity in the kidney. The total nitric oxide synthase activity is 25 times greater in the inner medulla than in the cortex, implicating greater frailty in the inner medulla [136]. Further, the activation of the complement system through each pathway (classical, alternative, leptin) was shown to play a significant role in cardiac ischemia/reperfusion injury both in mice and humans [121,137,138,139]. The activation of an alternative pathway was detectable during laparoscopy in humans [140]. The activation of both leptin and an alternative complementary pathway was reported to aggravate renal injury in mice [141]. 

### 6.3. Protective Drugs

Multiple drugs have an antioxidant effect, which can be protective during laparoscopy. Zinc and N-acetyl-cysteine were reported to be associated with an elevated level of catalase, while that of the superoxide dismutase was decreased [134,142]. The immediate beneficial effect of N-acetyl-cysteine has been debated recently since it might cause analytic interference with the measurement of creatinine [143]. Pentoxyfyllin increases the activity of the catalase enzyme without any adverse impacts on other enzymes [134,142]. The administration of these drugs resulted in more pronounced advantages compared to the pre- or post-conditioning methods. The protective effect of N-acetyl-cysteine against GFR drops was detectable even 72 h after 180 min of pneumo-peritoneum in rats [142,144,145]. Caffeic acid phenethyl ester, a component of the honey bee product propolis, capable of completely blocking xanthine oxidase and oxygen free radical production at a 10 µM concentration in vitro, proved also to be protective against oxidative stress caused by pneumo-peritoneum in rats [146]. 

The administration of a superoxide dismutase mimetic agent (tempol) was reported to increase medullary but not cortical blood flow by 16% [38]. This effect could be escalated further with the co-administration of catalase, whereas medullary perfusion fell, followed by the nitric oxide synthase inhibitor (L-NAME). These results suggest that H_2_O_2_ works as a vasoconstrictor, which can be one of the main determinants of the basal medullary vascular tone under physiologic conditions. H_2_O_2_ exerts its effect partly by the abolishing of NO-mediated vasodilatation.

From anesthetics, thiopental and propofol induction was associated with lower malondialdehyde concentrations during experimental kidney ischemia in rats, with similar effects on catalase activity [147]. 

## 7. Inflammatory Response

Multiple processes (NO, ischemic/reperfusion) show the involvement of the endothelial cells’ activity during laparoscopy. These are in close connection with each other and with the activation of pro-inflammatory mechanisms. From these, eicosanoid production, interleukins (ILs) and the insulin-like growth factor were investigated in the context the of laparoscopy.

### 7.1. Eicosanoids

Eicosanoids can result in either vasoconstriction (thromboxane A_2_, leukotrienes, hydroxy-eicosate-tranoic acid) or vasodilatation (prostaglandin I_2,_ known as prostacyclin, prostaglandin E_2_, epoxy-eicosatrienoic acids) [19]. The nonsteroidal analgesics block the cyclooxygenase pathway by weakening the vaso-dilatory potential.

### 7.2. Interleukin Family

Several cells in the kidney (podocytes, mesangial, endothelial, and tubular epithelial cells) can produce interleukin (IL)-6 [148]. Only podocytes express IL-6 receptors, but all listed cells contain the common signal transducer subunit of the receptor of the IL-6 cytokine family. Systemic effects, such as a 60-min bilateral renal ischemia, are associated with elevated IL-6 levels that lead to the generation of reactive oxygen species, endothelial dysfunction and vasoconstriction concluding in AKI [148,149]. 

In animals resuscitated from cardiac arrest, renal tissue impairment was far less extended than in those where the regional cessation of blood flow had been implemented [122]. In murine models, cytokine levels IL-1α, IL-1β, IL-2, IL-10, TNF- α, IFN-γ and monocyte chemoattractant protein-1 remained unchanged during and after the ischemic period. A rise in renal keratinocyte-derived chemokine, IL-6 and G-CSF was detected, but only the keratinocyte-derived chemokine demonstrated a difference between the groups after a 30-min ischemic period. During venous clamping, the level of keratinocyte-derived chemokine reached about 70% of those resulted from arterial obstruction, and these two effects were additive when the pedicle was clipped [122]. Serum IL-10 concentrations were significantly lower and IL-6 concentrations were significantly higher than in mice with cardiac arrest. To summarize these findings, even deteriorated venous outflow can lead to necrosis in each region of the kidney, partly via increased cytokine production, with a pattern different from whole-body ischemia.

The serum level of the pro-inflammatory cytokine IL-18 rises gradually with the increase of intraperitoneal pressure in 4-mmHg steps from 0 to 12 mmHg in rats, while insufflation time varies between 60 and 240 min [150]. This course is parallel with the elevation of serum AKI markers, like neutrophil gelatinase-associated lipocalin and cystatin-C. The contribution of lymphatic endothelial cells is not entirely clear, but a growing body of data supports their role both in the local and systematic clearance of chemokines [103]. 

### 7.3. Insulin-like Growth Factor 1

Insulin-like growth factor 1 (IGF-1) production is regulated growth hormones [151]. Other growth factors (epidermal, fibroblast, vascular epithelial, hepatocyte, platelet-derived and transforming growth factor β1) are also contributors of AKI [151,152]. Their decreased level is associated with programmed cell death and inflammatory processes, while their raised serum concentration is responsible for cell proliferation and fibrosis during transmission into chronic kidney disease [152,153]. The decrease of the serum IGF-1 concentration was significantly higher for the conventional small bowel resection group than in the laparoscopic group as represented in a rodent model. The serum IGF-1 concentration also returned to baseline earlier in the laparoscopic group [154]. 

## 8. The Effects of Anesthesia

Laparoscopy is performed usually under general anesthesia. In sheep experiments, RBF is reduced by 30–50% during general anesthesia alone; however, it is known to increase immediately after renal denervation [16,40,155,156]. RBF returned to the level of control animals within 5–13 days after surgical denervation, suggesting the existence of a possible escape mechanism [40]. Volatile anesthetics depress the firing (discharge) rate of baroreceptors and increase renal sympathetic nerve activity by removing the central nervous system inhibitory tone before the decline at high concentrations of agents [40,155,157,158]. The ratio between renal and aortic nerve activities varies among inhalational agents [158]. Furthermore, this effect seems to be at least partly counter-balanced by a hypotension-evoked increase of sympathetic tone [159]. Nitrous oxide produces the opposite effect on both MAP and renal sympathetic activity [157]. While neither angiotensin-convertase enzyme inhibitors nor angiotensin receptor blockers can exert any influence on RBF alone, this is significantly different under circumstances of systemic anesthesia.

The decrease of RBF during anesthesia can be diminished by the administration of enalapril in rabbits with or without mild-to-moderate congestive cardiac heart failure, but systematic hemodynamic responses become more pronounced [155]. Losartan, an AT_1_-receptor antagonist, has also been reported to improve RBF under isoflurane anesthesia in sheep, an effect that can be abolished by the administration of either the direct alfa-1 inhibitor prazosin or the vasopressin V_1_-receptor antagonist [160]. Alone or in combination, the latter two agents did not exert any effect on RBF.

Volatile anesthetics are known to attenuate the inflammatory response in murine models and exert a reno-protective effect in comparison with pentobarbital and ketamine [121,161]. Desflurane had a poorer performance than the other inhalative agents. It is not metabolized up to seven-fold minimal alveolar concentration for anesthesia [162]. Isoflurane and sevoflurane were shown to raise serum fluoride concentration, which can be nephrotoxic [163]. The reaction of sevoflurane and the CO_2_ absorber produces Compound A, a trifluoro methyl vinyl ether, which accumulates and exerts its renal injurious effect at low, minimal and metabolic flow anesthesia [162].

The volatile anesthetic methoxy-flurane is known as a definite nephrotoxic agent [164]. Methoxy-flurane and the other anesthetic gases (halothane, enflurane, isoflurane, sevoflurane, desflurane) are fluorinated ethers, and their degradation product, the inorganic fluorid, was thought to be responsible for nephrotoxicity [165]. This mechanism has not been detected yet in newer agents [166]. Sevoflurane reacts with the CO_2_-absorber creating halo-alkens (called as Compound A), which are severely deleterious for kidney function in rats, but not in humans. On the other hand, several reno-protective properties of inhalative anesthetics have been published: (1) diminished pro-inflammatory cytokine production caused by the trifluoro-carbon group of anesthetics; (2) increased release of the anti-inflammatory molecule transforming growth factor β1 from macrophages; (3) increased amounts of anti-apoptotic sphingosine-1-phosphate in cell membranes; (4) increased local adenosine production; and (5) enhanced IL-11 synthesis mitigates the effects of ischemic/reperfusion injury [165].

The exposure to inorganic fluorid can be eliminated by the administration of intravenous anesthetic agents. Propofol, a commonly used drug is found to be reno-protective, but the exact mechanism is still unclear. The upregulation of heme-oxygenase 1 expression can be involved, which promotes the conversion of heme to biliverdin, with antioxidant and anti-inflammatory carbon monoxide generation [121]. Propofol itself can have a scavenger property against ROS because of its phenol hydroxyl group. Dexmedetomidine, an α_2_-adrenoreceptor agonist was shown to have several advantageous influences against kidney injury: direct tubular effects, reducing renin levels, central inhibition of vasopressin, and the attenuation of ischemia/reperfusion injury [121]. The mechanism is not fully understood, although multiple pathways have been discovered.

## 9. The Effects of Retroperitoneal Insufflation

This technique can be applied only in certain types of surgery. Retroperitoneal insufflation has a theoretical advantage of having a lower CO_2_ load while peritoneal absorption is lacking, but it needs higher pressures to apply in animal models [113]. During the retroperitoneal approach, the level of serum aldosterone was 1.5 times higher compared to laparoscopy in pigs. After desufflation, it returned to below the baseline during laparoscopy, while retro-peritonoscopy accounted for only a slight decrease, resulting in doubled aldosterone concentrations after intervention. Abdominal wall lifting alone does not affect aldosterone levels significantly. Establishing pneumo-peritoneum with CO_2_ (but not with argon) was demonstrated to result in increased serum vasopressin levels and a proportional increase in the systemic vascular resistance of pigs [53]. This effect could be abolished by administering a vasopressin-antagonist. Serum osmolarity remained unchanged in each (CO_2_, argon) group. These authors emphasize the importance of avoiding the use of opiates during their experiments. Opiates are potential inhibitors of vasopressin’s neuro-hypo-physeal release but omitting them seems to be unethical even in preclinical conditions and makes it difficult to convert their results into human surgery.

## 10. Conclusions

In summary, the regulation of intra-renal flow may be regionally impaired and disconnected from overall RBF, in particular under circumstances of venous congestion or increased interstitial or external pressures. The widely accepted presence of renal arterial autoregulation refers only to cortical blood flow but is not fully applicable to medullary perfusion. A compromised venous outflow exerts its effect mainly through the impairing function and integrity of the renal medulla. Lymphatic drainage is difficult to assess but major reduction of it can potentially turn into an additive contributor to AKI. All these factors are affected by the implementation of pneumo-peritoneum generation during intra-abdominal laparoscopic surgery. Laparoscopy use has widely expanded during abdominal surgeries and the increasingly complex procedures carried out. Understanding the underlying pathologic processes and discovering the potential protective measurements could provide further refinement to minimize ischemic-reperfusion injury and the activation of the inflammatory processes. Further refinements are provided by the several humoral factors derived as a consequence of ischemic-reperfusion injury and the activation of the inflammatory process of anesthesia itself. Non-pharmacologic methods and several candidate drugs have the potential to diminish the detrimental effects of procedural pneumo-peritoneum formation or the anesthesia itself. Altogether, a better understanding of the relationship between pneumo-peritoneum formation and renal perfusion will support basic and clinical research, leading to improved clinical care and collaboration among medical specialties.

## Figures and Tables

**Figure 1 jcm-11-02728-f001:**
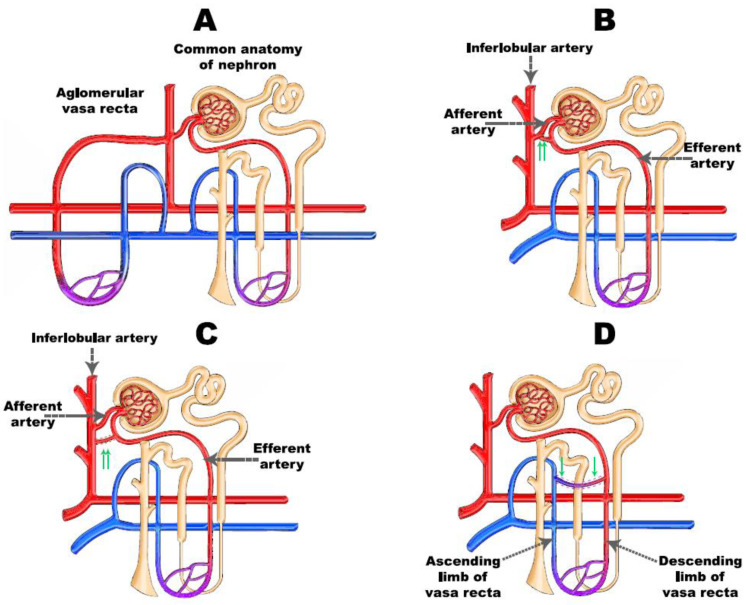
The pre-existing shunts and connections between renal vessels. (**A**) normal anatomy (right) and aglomerular vasa recta (left); (**B**) shunt between afferent and efferent arteries; (**C**) shunt between interlobular artery and vasa recta; (**D**) intra-medullar shunt in the vasa recta system. Red vessels: arteries; blue vessels: veins; purple vessels: connection vessels between arteries and veins; yellow tubes: urine conducting system; green arrows: irregular vessels.

**Figure 2 jcm-11-02728-f002:**
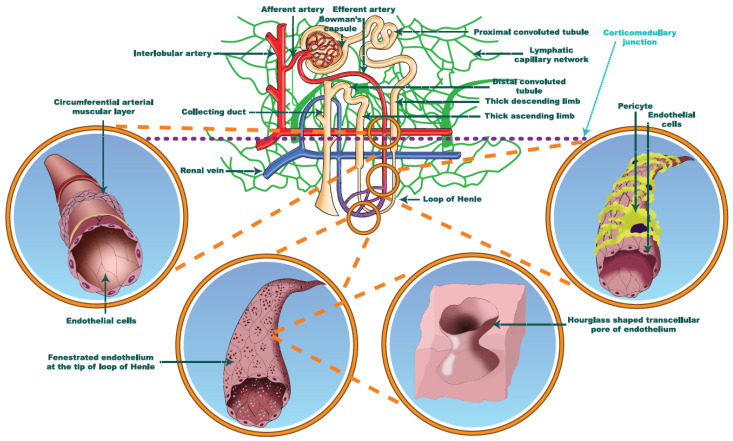
The microscopic anatomy of the arterial wall and the endothelium of vasa recta along their course towards the tip of medulla. Red vessels: arteries; blue vessels: veins; purple vessels: connection vessels between arteries and veins; yellow tubes: urine conducting system; green vessels: lymphatic network.

**Figure 3 jcm-11-02728-f003:**
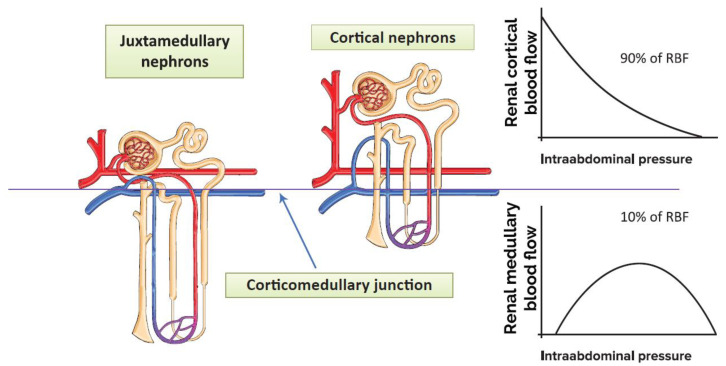
The different effects of elevated intra-abdominal pressure on cortical and medullary blood flow. Red vessels: arteries; blue vessels: veins; purple vessels: connection vessels between arteries and veins; yellow tubes: urine conducting system (Re-drawn with permission of Publisher, from [56]).

**Figure 4 jcm-11-02728-f004:**
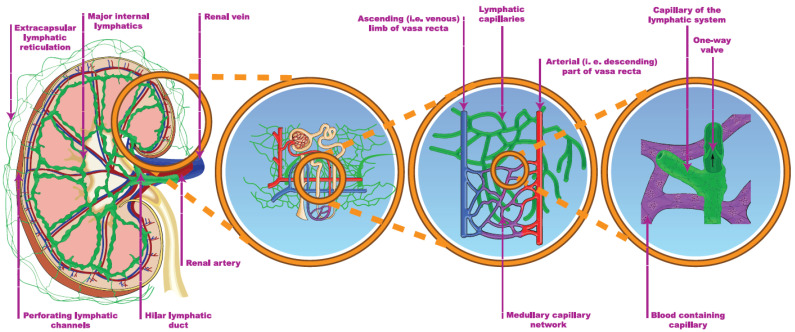
The renal lymphatic system (hilar and capsular). Red vessels: arteries; blue vessels: veins; purple vessels: connection vessels between arteries and veins; yellow tubes: urine conducting system; green vessels: lymphatic network.

**Figure 5 jcm-11-02728-f005:**
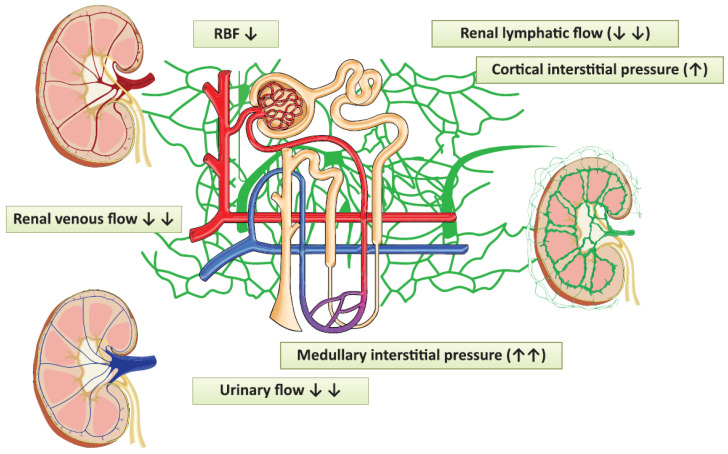
The pathophysiological changes evoked by laparoscopy, which can contribute to the development of acute kidney injury. Red vessels: arteries; blue vessels: veins; purple vessels: connection vessels between arteries and veins; yellow tubes: urine conducting system; green vessels: lymphatic network; RBF: renal blood flow.

## Data Availability

Not applicable.

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
