# Peer review of "Mediators of Regional Kidney Perfusion during Surgical Pneumo-Peritoneum Creation and the Risk of Acute Kidney Injury—A Review of Basic Physiology"

_jcm, 2022, doi:10.3390/jcm11102728_

Round 1

Reviewer 1 Report

I appreciate that you invited me to review the manuscript. In this review, the structure and the physiology in renal blood flow are well explained so that even the beginners could understand the basic physiology in kidney. In addition, the effect of intra-abdominal pressure on kidney is elucidated in the various mechanisms.

Author Response

Response to Reviewer 1 Comments

Point 1: I appreciate that you invited me to review the manuscript. In this review, the structure and the physiology in renal blood flow are well explained so that even the beginners could understand the basic physiology in kidney. In addition, the effect of intra-abdominal pressure on kidney is elucidated in the various mechanisms.

Response 1: Thank you for your kind opinion.

Reviewer 2 Report

In this paper Kopitkó et al. review the impact of laparoscopic surgery on kidney function. The paper is well structured and written, not only it is an excellent manuscript on the topic, but it provides also noticeable insights on renal physiology. Please find below my comments:

  • Overall, the manuscript provides great insight in the data: describes the used models, the observations from the original manuscripts and highlights also the possible limitations. It does, however, miss some clinical data. How relevant/frequent is the described issue? Are there any publication about AKI frequency and classification after laparoscopy? Additionally, some clinical studies reinforce the excellently explained physiology mechanisms (e.g., DoReMiFa study and fluid overload)
  • Lines 60-62: “The kidneys alone account for approximately 20% of cardiac output [at rest], a number that has been remarkably preserved across mammalian species. The renal blood flow (RBF) is surprisingly steady, but it has a significant circadian oscillation” This last part is contradictory, please explain better. Also notice the []
  • A suggestion about the introduction: a reminder about sterling forces and net ultrafiltration pressure together with the value of pressure in standard peritoneal inflation would help understanding the phenomenon.
  • Figure 1: the letter order is wrong.
  • Lines 111-113 refer to Fig 2, but I cannot relate them
  • Line 123 “they wrapped changes around.” Is most likely an erroneous editing
  • Line 160 consider: it “may result” rather than “results”
  • Line 186: when referring to peritoneal inflation in animals (here rats) might be of interest to provide the animal net ultrafiltration pressure
  • Line 198-199: I’d refer here to Fig 2.
  • Line 231: “clinically important significance.” Is redundant
  • Lines 242-248 about the splanchnic reserve: how does this connect with the article?
  • Line 263: “so long” change to “as long”
  • Line 282: “Unexpectedly, venous occlusion leads to a greater damage in RBF than the cessation of arterial inflow.” I would think than any occlusion abrogates 100% of the flow. Maybe same delta-pressure induce different effect in the 2 compartments?
  • Line 427 see []: The widely known parts of the renin-angiotensin-aldosterone system (RAAS) are 427 renin, angiotensinogen, the angiotensin converting enzyme and angiotensin II [and aldosterone].
  • The whole RAAS chapter is barely connected with the article focus
  • Line 462: why aldosterone changes in laparoscopic surgery are mentioned in the TG feedback chapter?
  • Line 594: NAC might actually falsely improve kidney function by perturbating creatinine measures (https://www.ncbi.nlm.nih.gov/pmc/articles/PMC6156188/)

Author Response

Response to Reviewer 2 Comments

Point 1: Overall, the manuscript provides great insight in the data: describes the used models, the observations from the original manuscripts and highlights also the possible limitations. It does, however, miss some clinical data. How relevant/frequent is the described issue? Are there any publication about AKI frequency and classification after laparoscopy? Additionally, some clinical studies reinforce the excellently explained physiology mechanisms (e.g., DoReMiFa study and fluid overload)

Response 1: Generaly speaking incidence of kidney injury ranges between 2 and 41% after major intraabdominal surgery. Epidemiological data of laparoscopy were reviewed by Demyttenaere et al (Demyttenaere, S., Feldman, L.S. & Fried, G.M. Effect of pneumoperitoneum on renal perfusion and function: A systematic review. Surg Endosc 21, 152–160 (2007). https://doi.org/10.1007/s00464-006-0250-x), but still before the first standardized definition of AKI (RIFLE, 2004). That is why hard to provide precise data, although perioperative AKI is thought to be underdiagnosed (Meersch M, Schmidt C, Zarbock A. Perioperative Acute Kidney Injury: An Under-Recognized Problem. Anesth Analg. 2017 Oct;125(4):1223-1232. doi: 10.1213/ANE.0000000000002369. PMID: 28787339.; https://clinicaltrials.gov/ct2/show/NCT04755452). The diagnostic criteria of AKI are still a bit uncertain (AKIN, KDIGO, biomarkers, urine output, renal surgery etc.; https://journals.plos.org/plosone/article?id=10.1371/journal.pone.0247088). We collected the clinical data also, but it resulted as voluminous text as this manuscript is. We are planning to publish it in the near future.

Point 2: Lines 60-62: “The kidneys alone account for approximately 20% of cardiac output [at rest], a number that has been remarkably preserved across mammalian species. The renal blood flow (RBF) is surprisingly steady, but it has a significant circadian oscillation” This last part is contradictory, please explain better. Also notice the []

Response 2: Thank you for your opinion. The text was rearranged according to your suggestions, as follows (Line 75): “The kidneys alone account for approximately 20% of cardiac output uptake at rest, a number that has been remarkably preserved across mammalian species. As it is widely known, renal autoregulation keeps the renal plasma flow (RPF) stable under physiological conditions, when the mean arterial pressure (MAP) is maintained between 80 and 160 mmHg, mirroring the balance between the afferent and efferent arteriolar tones in the cortex. 18,19 The renal blood flow (RBF) has a significant circadian oscillation: the peak value can reach 150% of the nadir RBF. 20,21

Point 3: A suggestion about the introduction: a reminder about sterling forces and net ultrafiltration pressure together with the value of pressure in standard peritoneal inflation would help understanding the phenomenon.

Response 3: Thank you for your suggestion. A paragraph has been inserted into the introduction (Line 51): “The transendothelial fluid shift is governed by the hydrostatic and colloid osmotic forces of both side of glycocalyx according to Starling’s principle.14 The net ultrafiltration pressure is derived from the opposing forces of the glomerular blood hydrostatic pressure (approximately 55 mmHg under physiological condition in humans) vs the combinations of capsular hydrostatic pressure (about 15 mmHg) and blood colloid osmotic pressure (about 30 mmHg).15 With an assumption of unchanged capillary permeability and surface area and an ultrafiltrate that is practically protein-free without colloid osmotic pressure, the net ultrafiltration pressure calculates to around 10 mmHg only, a surprisingly low number.”

Point 4: Figure 1: the letter order is wrong.

Response 4: Thank you, for your opition. The order of letters has been corrected.

Point 5: Lines 111-113 refer to Fig 2, but I cannot relate them

Response 5: Thank you for your opinion. Fig 2 has been deleted from here and moved to line 219. Fig 3 has been renumbered as Fig 2 as a consequence.

Point 6: Line 123 “they wrapped changes around.” Is most likely an erroneous editing

Response 6: Thank you for your opinion. The erroneous editing was corrected as “they wrapped around.”

Point 7: Line 160 consider: it “may result” rather than “results”

Response 7: Thank you for your opinion. The text has been corrected according to your suggestions.

Point 8: Line 186: when referring to peritoneal inflation in animals (here rats) might be of interest to provide the animal net ultrafiltration pressure

Response 8: Thank you for your suggestion. The net ultrafiltration pressure was reported as 14 mmHg, which is very similar to human (10 mmHg). This sentence has been inserted into the text with ref ((Line 208):

“The net ultrafiltration pressure was reported as 14 mmHg, which is very similar to the one calculated in human (10 mmHg).52

Point 9: Line 198-199: I’d refer here to Fig 2.

Response 9: Thank you for your suggestion. Fig 2 has been moved to here.

Point 10: Line 231: “clinically important significance.” Is redundant

Response 10: Thank you for your opinion. The expression has been simplified (Line 252).

Point 11: Lines 242-248 about the splanchnic reserve: how does this connect with the article?

Response 11: Thank you for your question. The splanchnic reserve volume is significant proportion of blood volume. During establishing pneumoperitoneum it can be drained into circulating volume increasing venous return and consequently the cardiac output – having even more volumen effect, if it is diluted by extravascular fluids to normal hematocrit. On the other side, increased IAP can cause renal venous congestion even though kidney is retroperitoneal organ, which is emphasized at this point. The opposite processes can be observed after desufflation of pneumoperitoneum. The regulatory nervous and humoral factors can influence these changes.

The text has been modified as follows (Line 265-269): “It can be pushed into the systemic circulation when the sympathetic nervous system activated resulting a significant intravascular volume expanding effect when the red blood cells are diluted up to physiologic level.64 Augmented venous return to the right atrium may lead into increased cardiac output, but on the other side, increased IAP can cause renal venous congestion.”.

Point 12: Line 263: “so long” change to “as long”

Response 12: Thank you for your opinion. The text has been corrected according to your suggestions (Line 288).

Point 13: Line 282: “Unexpectedly, venous occlusion leads to a greater damage in RBF than the cessation of arterial inflow.” I would think than any occlusion abrogates 100% of the flow. Maybe same delta-pressure induce different effect in the 2 compartments?

Response 13: Thank you for your opinion. The text has been modified according to your suggestions Line 307): “Unexpectedly, hampered venous outflow leads to a greater damage in RBF than the cessation of arterial inflow.”

Point 14: Line 427 see []: The widely known parts of the renin-angiotensin-aldosterone system (RAAS) are 427 renin, angiotensinogen, the angiotensin converting enzyme and angiotensin II [and aldosterone].

Response 14: Thank you for your opinion. The text has been corrected according to your suggestions, as follows (Line 452): “The renin-angiotensin-aldosterone system (RAAS) is one of the main contributors of renal macro- and microhemodynamics. Its widely known parts are renin, angiotensinogen, the angiotensin converting enzyme, angiotensin II and aldosterone.”

Point 15: The whole RAAS chapter is barely connected with the article focus

Response 15: Thank you for your opinion. The text has been corrected according to your suggestions (Line 482):

“The myogenic mechanism of afferent arterioles and the tubulo-glomerular feedback are the main contributors of RBF besides sympathetic innervation.19 A so-called third mechanism (3M) also exist representing vasoconstrictive forces acting on afferent glomerular arteries with slower reaction time. The physiological background of 3M is uncertain, but some argument supports the possible role of RAAS.116

Point 16: Line 462: why aldosterone changes in laparoscopic surgery are mentioned in the TG feedback chapter?

Response 16: The TG mechanism is influenced by RAAS as it is mentioned in the manuscript. To show a more holistic picture, a short sentence about of third mechanism of glomerular hemodynamics (Line 482).

“The myogenic mechanism of afferent arterioles and the tubulo-glomerular feedback are the main contributors of RBF besides sympathetic innervation.19 A so-called third mechanism (3M) also exist representing vasoconstrictive forces acting on afferent glomerular arteries with slower reaction time. The physiological background of 3M is uncertain, but some argument supports the possible role of RAAS.116

Point 17: Line 594: NAC might actually falsely improve kidney function by perturbating creatinine measures (https://www.ncbi.nlm.nih.gov/pmc/articles/PMC6156188/)

Response 17: Thank you for your opinion. The text has been corrected as follows Line 620):

“The immediate beneficial effect of N-acetylcysteine was debated recently since it might can cause analytic interference with the measurement of creatinine.147

Reviewer 3 Report

Very well written review with excellent illustrations. It really brings up a good area for discussion and awareness. While the manuscript is very comprehensive, a small paragraph about future direction in this area and how it can be expanded upon, will be a nice addition to the conclusion segment. 

Author Response

Response to Reviewer 3 Comments

Point 1: Very well written review with excellent illustrations. It really brings up a good area for discussion and awareness. While the manuscript is very comprehensive, a small paragraph about future direction in this area and how it can be expanded upon, will be a nice addition to the conclusion segment. 

Response 1: Thank you for your kind opinion. A few sentences were added to conclusion section, as follows (Line 766): “Laparoscopy use have widely expanded during abdominal surgeries and increasingly complex procedures carried out. Understanding the underlying pathologic processes and discovering the potential protective measurements could provide further refinement to minimize ischemic-reperfusion injury and the activation of the inflammatory processes. Further refinements are provided by the several humoral factors derived as a consequence of ischemic-reperfusion injury and the activation of the inflammatory process or anesthesia itself. Non-pharmacologic methods and several candidate drugs have the potential to diminish the detrimental effects of procedural pneumoperitoneum formation or the anesthesia itself. Altogether, a better understanding of the relationship between pneumoperitoneum formation and renal perfusion will support basic and clinical research, leading to improved clinical care and collaboration among medical specialties.”